# The Agreement between Wearable Sensors and Force Plates for the Analysis of Stride Time Variability

**DOI:** 10.3390/s24113378

**Published:** 2024-05-24

**Authors:** Patrick Slattery, L. Eduardo Cofré Lizama, Jon Wheat, Paul Gastin, Ben Dascombe, Kane Middleton

**Affiliations:** 1Sport, Performance and Nutrition Research Group, School of Allied Health, Human Services and Sport, La Trobe University, Melbourne, VIC 3083, Australia; p.slattery@latrobe.edu.au (P.S.); eduardo.cofre@unimelb.edu.au (L.E.C.L.); p.gastin@latrobe.edu.au (P.G.); 2Department of Nursing and Allied Health, School of Health Sciences, Swinburne University of Technology, Hawthorn, VIC 3122, Australia; 3Department of Medicine, The University of Melbourne, Parkville, VIC 3050, Australia; 4Academy of Sport and Physical Activity, Sheffield Hallam University, Sheffield S10 2DN, UK; jon.wheat@ntu.ac.uk; 5School of Science and Technology, Nottingham Trent University, Nottingham NG11 8NS, UK; 6Applied Sport Science and Exercise Testing Laboratory, School of Life and Environmental Sciences, University of Newcastle, Ourimbah, NSW 2258, Australia; b.dascombe@westernsydney.edu.au; 7Sports and Exercise Science, School of Health Sciences, Western Sydney University, Sydney, NSW 2000, Australia

**Keywords:** variability, walking, regularity, stride, load carriage, sensors

## Abstract

The variability and regularity of stride time may help identify individuals at a greater risk of injury during military load carriage. Wearable sensors could provide a cost-effective, portable solution for recording these measures, but establishing their validity is necessary. This study aimed to determine the agreement of several measures of stride time variability across five wearable sensors (Opal APDM, Vicon Blue Trident, Axivity, Plantiga, Xsens DOT) and force plates during military load carriage. Nineteen Australian Army trainee soldiers (age: 24.8 ± 5.3 years, height: 1.77 ± 0.09 m, body mass: 79.5 ± 15.2 kg, service: 1.7 ± 1.7 years) completed three 12-min walking trials on an instrumented treadmill at 5.5 km/h, carrying 23 kg of an external load. Simultaneously, 512 stride time intervals were identified from treadmill-embedded force plates and each sensor where linear (standard deviation and coefficient of variation) and non-linear (detrended fluctuation analysis and sample entropy) measures were obtained. Sensor and force plate agreement was evaluated using Pearson’s r and intraclass correlation coefficients. All sensors had at least moderate agreement (ICC > 0.5) and a strong positive correlation (r > 0.5). These results suggest wearable devices could be employed to quantify linear and non-linear measures of stride time variability during military load carriage.

## 1. Introduction

Injury in the military is problematic, as it impacts a soldier’s physical and mental health, training, and progression and an army’s general readiness for deployment [1]. The burden of injury is predicted to cost the Australian Defence Force more than AUD 210 million per year, with around 17.7 injuries occurring per 100 person-years of active service [2,3]. Load carriage has become a focal point of investigation as one of the highest self-reported causes of soldier injury and has been shown to contribute to overall musculoskeletal injury (MSI) risk [4,5,6]. Military load carriage commonly exceeds 20 kg during general patrol duties; however, evidence suggests that >13 kg of a load can increase MSI risk by 50–60% [7]. At a macro level, applying load management principles such as adjusting the intensity, frequency, and duration of military training activities can decrease the number and severity of injuries [8,9]. However, much research has endeavoured to identify the more subtle predispositions, behaviours, adaptions, and risk factors that precede military-related injury.

The examination of gait characteristics under military-relevant load has shown that stride length, stride frequency, gait variability, and trunk lean are all affected when compared with unloaded gait [10,11]. Kinematic changes may be necessary, but understanding their potential thresholds and identifying maladaptation could be vital to understanding injury risk. Traditional linear statistical measures of stride time variability, such as the standard deviation (SD) and coefficient of variation (CV), are used to quantify the magnitude of stride-to-stride fluctuations [12]. These linear measures have primarily been used in the aging population and can differentiate between a healthy gait, those at an increased fall risk [13], and those with motor diseases [14]. Regarding military load carriage, Springer [15] performed a longitudinal study monitoring the gait characteristics of 76 soldiers during their first year of service. Using multiple logistic regression, they reported that linear stride time variability was associated with lower-body injury risk. Despite this, non-linear measures are considered more sensitive than linear measures [16] but have yet to be used to examine stride time variability during military load carriage in the context of injury risk. Further information exists within the temporal structure of stride time and can be explored using non-linear measures such as detrended fluctuation analysis (DFA) [17,18] and sample entropy [19]. DFA quantifies the persistence of stride time patterns, whereas SE measures the predictability or regularity. A healthy stride time pattern exhibits persistent long-term correlation; on the contrary, a loss of persistence and an increase in regularity are often associated with an impaired gait [20]. Although non-linear measures are more sensitive to changes in stride time than linear measures, they require the accurate detection of heel contact events.

Force plates are considered one of the gold standard methods for recording heel contacts [21,22]. However, these can be large, expensive, and immobile, which limits the application and accessibility of stride time analysis in a field setting [23]. Wearable devices (e.g., inertial measurement units (IMUs)) offer a potential solution, as they are comparably smaller, lighter, and inexpensive, making them ideal for monitoring soldiers during military activities [24]. Previous research has shown that IMUs attached to the foot are valid for detecting heel contact events, stride time, walking speed, swing time, stride length, and cadence [25,26]. However, IMU validity for measuring linear variability has been reported to range from poor to excellent (ICC: 0.22–0.90) when sensors are placed at the ankle or waist [27,28]. The difficulty of consistently identifying heel contact events from the IMU signal over time is thought to contribute to the validity [28,29]. Few studies have assessed the use of IMUs for quantifying non-linear measures of stride time, such as DFA [30]. To date, no studies have explored these measures in a military context, where speeds and loads can affect a soldier’s walking dynamic behaviour.

The aim of this study was to assess the agreement between several wearable sensors and force plates to quantify linear and non-linear stride time measures using recommended approaches [31]. The results will help determine the utility of wearable sensors in measuring non-linear stride time variables of walking in the military context, which can be potentially used as markers of MSI risk.

## 2. Methods

### 2.1. Participants

Nineteen Australian Army trainee soldiers (6 females, 13 males, age: 24.8 ± 5.3 years, height: 1.77 ± 0.09 m, body mass: 79.5 ± 15.2 kg, service: 1.7 ± 1.7 years) participated in the study. All participants were free of neuromusculoskeletal injury for six months before participation. All participants provided written, informed consent to ethical procedures, which were approved by the Departments of Defence and Veterans’ Affairs Human Research Ethics Committee (Ethics protocol #302-20) and reciprocally approved by La Trobe University’s Human Ethics Committee (Ethics protocol #302-20 DDVA HREC). A sensitivity power analysis was conducted using G*Power (V 3.1.9.6, Keil, Germany [32]). With 19 participants and an α error probability of 0.05, 80% power would be achieved for r ≥ 0.32 (η^2^p = 0.1).

### 2.2. Experimental Overview

The research design of this study was part of a broader investigation aimed at exploring various biomechanical and physiological aspects of load carriage [33]. The participants attended three sessions, with each session taking place one week apart. In the first session, the participants completed two minutes of walking on a Tandem Force-Sensing Treadmill (AMTI Inc., Watertown, MA, USA) to familiarise themselves with the device and task. In each session, the participants completed a 12 min treadmill walk whilst holding a replica F88 Austeyr rifle (3.2 kg) at 5.5 km/h at a 0° incline. The participants were fitted with 23 kg of a body-borne load distributed via a weighted vest (anterior load: 17 kg, posterior load: 6 kg), and they wore loose active clothing with military boots (Figure 1).

The participants were instructed to walk in the middle of the treadmill and make an exaggerated stomp as their first step to assist with force plate and sensor synchronisation. A countdown of “3-2-1-go” was used to begin each trial. During the trials, the participants were informed about the remaining time, but no additional audible or visual stimulus was present throughout the trials. Five wearable sensors were secured to/in each participant’s boots using a combination of Fixomull, rigid tape, and/or Velcro straps (Figure 2). As this study was a part of a more extensive research programme, the participants also wore a portable spiroergometer, wireless electromyography sensors, and motion capture markers throughout their trials.

### 2.3. Data Analysis

#### 2.3.1. Instrumented Treadmill

The force and centre of pressure data were collected from the treadmill’s integrated force plates (FP; 1000 Hz) via Vicon Nexus software (v2.14, Vicon Motion Systems Ltd., Oxford, UK) and imported into MATLAB 2021a (Natick, MA, USA) [34] for post-processing. Time-series data were downsampled to 120 Hz using cubic spline interpolation to match the lowest sampling rate of the wearable sensors. A 120 Hz sample rate was deemed appropriate, as it met the suggested minimum frequency for capturing subtle variations in non-linear measures [35]. The treadmill featured dual tandem force plates, allowing a participant’s foot to move freely across both force plates while walking. Centre of pressure (COP) data were filtered using a 50 Hz second-order low-pass Butterworth filter. To calculate heel contact times, the antero-posterior (AP) COP was utilised. Heel contact was defined as the moment when the AP COP transitioned from a posterior to an anterior direction. MATLAB’s “findpeaks” function identified this local maximum for each contact. Medio-lateral (ML) COP was used to determine whether the first heel contact was from the left or right heel using the first three seconds of a trial, with the “findpeaks” function identifying the first occurrence.

#### 2.3.2. Wearable Sensors

Five sensors were used in this study: Axivity AX3 (AX; 200 Hz; Axivity Ltd., Newcastle upon Tyne, UK), Xsens DOT (XS; 120 Hz; Movella Inc., Henderson, NV, USA), Plantiga (PG; 500 Hz; Plantiga Technologies Inc., Vancouver, BC, Canada), Opal APDM (OP; 128 Hz; APDM Inc., Portland, OR, USA), and Vicon Blue Trident (BT; 1000 Hz; Vicon Motion Systems Ltd., Oxford, UK). The placement of the sensors on the boot is illustrated in Figure 2. All sensors were chosen due to them being commercially available, with the aim to include a sample of popular sensors that incorporated both cost-effective and high-end options. Each sensor recorded 3D-acceleration data, which was then saved in their respective proprietary software. The raw data were later exported into MATLAB 2021a for post-processing. Time series data from sensors with a sampling rate greater than 120 Hz were downsampled to 120 Hz using cubic spline interpolation. A 60 Hz second-order low-pass Butterworth filter was applied to the vertical acceleration. To commence, heel contacts were detected using MATLAB’s “findpeaks” function. The reference peak (RP) was calculated as the average peak height of all the peaks identified. Peaks exceeding half of the RP were accepted as heel contacts. In conjunction with the RP, the average time between accepted heel contacts was calculated and used to reanalyse the time series and find all heel contacts. The AX, XS, PG, and OP sensors captured right heel contacts, whereas the BT sensor captured left heel contacts.

### 2.4. Data Processing

Data recording began as the participant stood in a static position on the treadmill. BT data were synchronised with the treadmill force plates and recorded in Vicon Nexus. The first 20 s of recorded data were removed to eliminate any effects of gait initiation. Matched stride time series were then adjusted to include 513 consecutive heel contacts (512 stride times). The stride time series’ mean (M), CV, and SD were calculated in RStudio (RStudio, 2022 [36]), whereas the DFA and SE were calculated in MATLAB. The DFA-alpha scaling exponent was calculated using the average evenly spaced windows method, which has been shown to increase DFA-alpha’s precision [17]. The window size range was set from nmin = 4 to nmax = N/9 (57), with k (21) estimated using the method described by Liddy and Haddad [37]. SE was calculated following Richman and Moorman’s [38] method, with the parameters of r = 0.2 (tolerance ratio) and m = 2 (vector length).

### 2.5. Statistical Analysis

Out of a possible total of 1995 (participants (19), sensors (7, including left and right FP), sessions (3), stride time measures (5, including mean)) stride time data points, 1540 data points were collected from the participants. There were a few reasons for the reduced number of data points. One participant did not return after their initial session due to an unrelated injury, and several participants did not complete all conditions. Of the collected data points, 24 data points were identified as outliers that fell below −3.5 or above 3.5 using modified *z*-score cut-offs and were removed [39], leaving 1516 stride time data points. After this procedure, all variables were normally distributed (Shapiro-Wilk’s test) with homogeneous variance (Levene’s test; *p* > 0.05). Using the “irr” RStudio package [40], Pearson’s *r* was used to assess the relative agreement between the sensor and FP measures. A two-way random-effects single-measures intraclass-correlation model (ICC 2,1) was used to evaluate the absolute agreement. The agreement was categorised as poor (ICC < 0.5), moderate (ICC = 0.5–0.75), good (ICC = 0.75–0.90), or excellent (ICC > 0.9) using the criteria proposed by Koo and Li [41]. The significance was set at *p* < 0.05. Bland–Altman plots were produced using the “blandr” RStudio package [42]. The “Metrics” RStudio package [43] was used to calculate the root mean squared error (RMSE) between the measures obtained with each sensor and the FP. A paired-samples *t* test was performed using the “stats” RStudio package [36] to assess if the means between the sensors and the FP were statistically different for each measure.

## 3. Results

Table 1 presents summary statistics (mean, standard deviation, and RMSE) for the force plates (FP) and each wearable sensor for all variables. There were significant differences between the force plates and AX for the stride time mean and standard deviation, XS for the stride time mean, and APDM for all variables (Table 1).

### 3.1. Absolute Agreement

Absolute agreement between the sensor and FP-calculated stride time measures, as measured by ICC, was demonstrated to be at least moderate (*p* < 0.05; Table 2). Linear measures of SD and CV exhibited good-to-excellent agreement between all sensors and FP (ICC > 0.88). For the non-linear measures of DFA and SE, there was good-to-excellent agreement between FP and XS and AX and PG, whereas for BT and OP, the level of agreement was moderate-to-good (Table 2).

### 3.2. Relative Agreement

All sensors demonstrated significant (*p* < 0.05) and strong (*r* > 0.5) positive correlations with FP for all stride time measures (Table 3). Linear measures reported stronger correlations (*r* = 0.88–0.99) than non-linear measures (*r* = 0.65–0.91). The inspection of the Bland–Altman plots shows positive and negative dispersion for all sensors with the force plates, with very little evidence of systematic or proportional bias and very few data points outside the limits of agreement.

## 4. Discussion

This study aimed to evaluate the agreement between several wearable sensors and force plates to quantify linear and non-linear stride time measures. All sensors exhibited at least moderate agreement and a strong positive correlation with FP across all measures. Wearable sensors appear to be a cost-effective and portable option for assessing linear and non-linear measures of stride time and its variability during load carriage.

Linear measures (SD, CV) demonstrated good-to-excellent agreement (*ICC* > 0.85) and a strong positive correlation (*r* > 0.88) with FP. This is consistent with the ICC values reported for waist-worn IMUs [28] but demonstrated higher agreement than the reported values for IMUs placed on the ankle [44]. The observed differences are likely due to the dissimilar algorithms utilised for calculating heel strikes from the IMUs. Rantalainen [44] in 2019 used IMUs placed on the ankle and hypothesised that their poor concurrent validity resulted from their gyroscope-based heel detection algorithm, which is different from the accelerometer-based algorithm used in the current study. Rantalainen [28] was able to achieve excellent concurrent validity for waist-worn IMUs; however, this was achieved using an algorithm that removed heel strikes outside a ± 0.2-s threshold of the force-plate-recorded heel strike. The current study shows that linear measures calculated from wearable devices are comparable to those of force plates without the use of an algorithm that relies on another concurrent analysis.

The non-linear measures obtained with the wearable sensors exhibited lower agreement (*ICC* = 0.57–0.92) and weaker correlations (*r* = 0.65–0.91) with the FP than the linear measures. This is likely because non-linear measures explore the temporal structure of a time series rather than the overall magnitude of variability (e.g., SD), making them more sensitive to subtle changes occurring on a stride-by-stride basis. DFA and SE have a greater dependency on the accurate detection of gait events and are likely influenced by the disparity between the original sampling frequencies of the sensors and the force plates [45]. Liddy [46] explored the effect of the sampling rate on stride time DFA alpha and concluded that a lower sampling frequency could distort a stride time series and decrease the presence of long-term correlations. However, the latter only occurred for sampling frequencies below 120 Hz, with DFA calculations using 120–360 Hz being unaffected. Notably, most of the included sensors in this study sampled within the mentioned range, except for PG (500 Hz) and BT (1000 Hz), which sampled at a higher rate. However, the FP recorded considerably higher values (1000 Hz) than most of the sensors, making FP more sensitive to subtle stride time variations. Alternatively, it is possible that using a sampling frequency that is too high may introduce non-biological white noise, which may result in lower agreement between the sensors and FP for non-linear measures [47].

It has been suggested that downsampling COP data can lead to a linear decrease in the DFA alpha and a linear increase in SE. This has the potential to affect the results of the current study, as several sensors and FPs were downsampled by different magnitudes to reach the common frequency of 120 Hz [48]. However, after conducting a subsequent comparison between the FP and the sensors using their original sampling frequency, the ICC and correlation values were similar when compared with the 120 Hz comparison (see Appendix A). The one exception to this was SE (decreased agreement with the original frequency), which is known to be affected by the sample frequency, as the interval between consecutive data points decreases when the sample rate is increased [49].

The Bland–Altman plots (Figure 3 and Figure 4) and the RMSE values (Table 1) demonstrate that there were only small measurement differences between each sensor and the FP when calculating linear and non-linear measures. Despite the generally small differences observed, there are instances where values fall outside the limits of agreement in the Bland–Altman plots (Figure 3 and Figure 4). These outliers may be due to measurement or calibration errors. Alternatively, they could be due to unexpected changes in a participant’s gait, which could affect their heel contact with the force plate and impact their detection by the sensor’s algorithm to a different extent. Furthermore, when comparing non-linear measures (Figure 4) to linear measures (Figure 3), there appears to be a greater dispersion in agreement, indicating more variability in agreement. The results of the paired *t* tests indicated a significant difference in the means calculated from certain sensors (OP, AX [SD]) when compared to the measurements obtained from the FP. Previous studies have shown that differences of >0.19 in DFA [50] and >0.66 in SE [51] can differentiate between healthy and impaired walking. Therefore, it is highly probable that wearable sensors could also be used to discriminate between similar population groups.

There were a few limitations that may have influenced the results of the study. The position of the OP sensors being over the shoelaces rather than close to the heel may have affected the stride time values obtained from this sensor, as demonstrated by the significant *t* test and lower levels of absolute and relative agreement. However, due to the OP software (Moveo Explorer, 2020 [52]) and sensor shape, it was not feasible to change its location. A similar issue occurred with the PG sensor, which only had one possible location (insole). Future research should randomise each device’s location to ensure that the sensor’s position on the foot does not affect the agreement with force plate measures. The decision to position the sensors around the boot of the participants was based on the consideration that this area would have a minimal impact on a soldier’s ability to perform their duties in a field setting. Consistent heel contact detection at the ankle/foot has been recognised as a potential limitation of existing gait analysis methods [26,27]. When quantifying a discrete signal (stride time) from a continuous signal, the discretisation error is present in the calculation of all measures. Although the sampling frequency may have affected the results of the present study, the aim of the study was to compare currently available technologies and not to explore the fundamentals of time series non-linear analysis and the effects of technical specifications on them. Each sensor’s absolute and relative agreement with the force plates sampling at their original frequency is shown in Appendix A (without the removal of outliers). All results are comparable to that of their downsampled agreement, apart from SE, which, as previously mentioned, is known to be influenced by sample frequency [49]). Continuous advancements in wearable devices and a consensus on the technical specifications and parameters used for non-linear analyses may help solve some of the present clinometric issues that prevent their expanded use.

In summary, the study demonstrated that linear and non-linear measures obtained with wearable devices’ data are comparable to those obtained with FP during a loaded military march. Therefore, this technology can be used to examine the potential use of dynamic behaviour metrics in future research, such as assessing injury risk, and should encourage broader and in-field research to understand how constraints (e.g., load, terrain) affect soldiers’ dynamic performance during military walking.

## Figures and Tables

**Figure 1 sensors-24-03378-f001:**
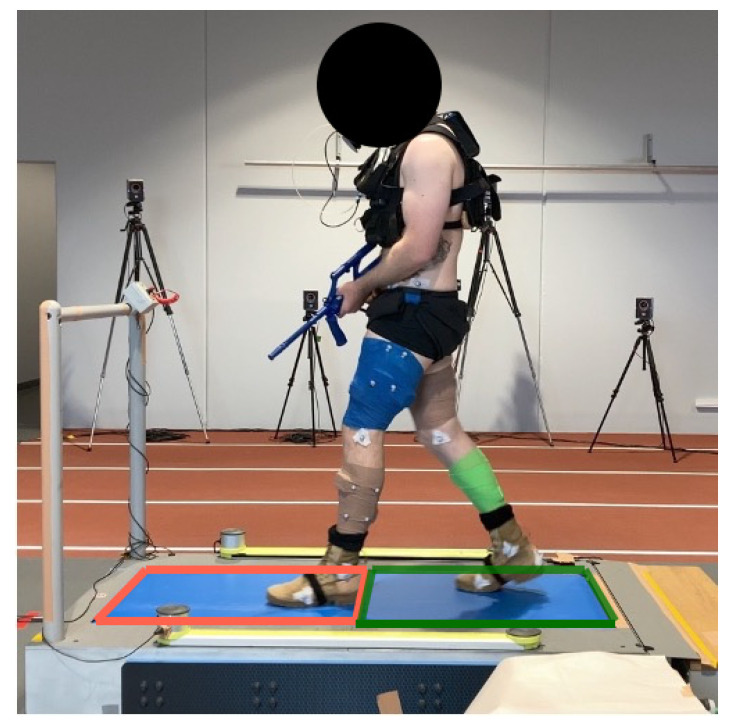
Experimental setup showing a participant on the instrumented treadmill. The coloured squares indicate the position of each embedded force plate.

**Figure 2 sensors-24-03378-f002:**
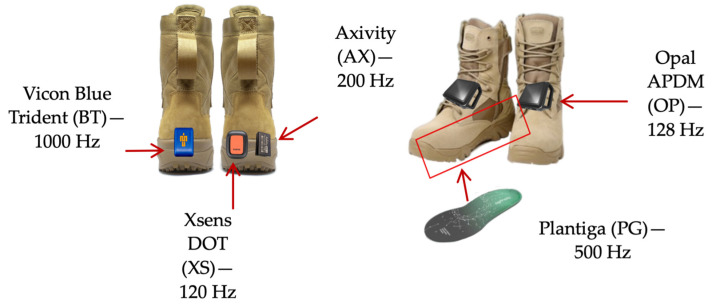
Schematic of sensor positions on/in military boots.

**Figure 3 sensors-24-03378-f003:**
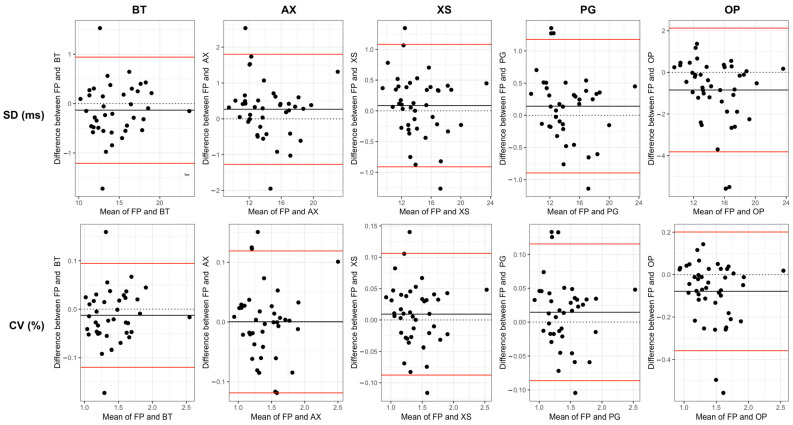
Bland–Altman plots for linear measures of stride time variability. Vicon Blue Trident (BT), Axivity (AX), Xsens Dot (XS), Plantiga (PG), Opal APDM (OP), Force Plates (FP).

**Figure 4 sensors-24-03378-f004:**
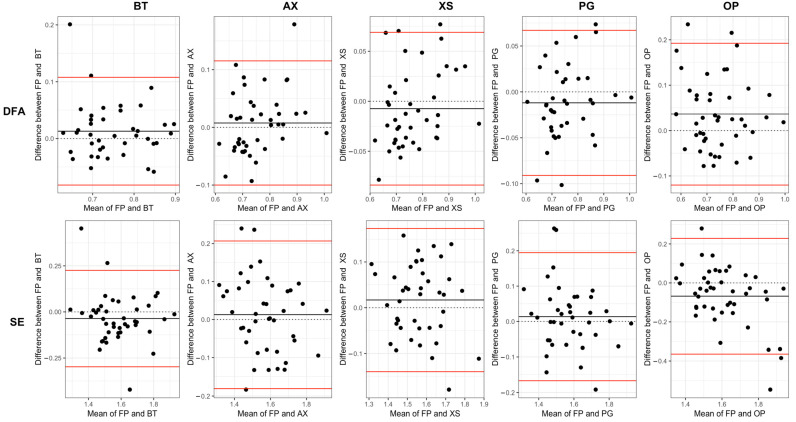
Bland–Altman plots for non-linear measures of stride time variability. Vicon Blue Trident (BT), Axivity (AX), Xsens Dot (XS), Plantiga (PG), Opal APDM (OP), Force Plates (FP).

**Table 1 sensors-24-03378-t001:** Summary statistics (Mean ± SD, RMSE) of stride time variables for the force plates (FP) and wearable sensors: Vicon Blue Trident (BT; left foot only), Axivity (AX), Xsens Dot (XS), Plantiga (PG), Opal APDM (OP). Statistically significant differences between the sensor and the FP are denoted by * *p* < 0.005, ^+^
*p* < 0.001.

Measure	FP (Left)	FP (Right)	BT	AX	XS	PG	OP
Mean (s)	1.03 ± 0.05	1.03 ± 0.05	1.03 ± 0.05	1.01 ± 0.05 ^+^	1.03 ± 0.05 ^+^	1.03 ± 0.05	1.04 ± 0.05 ^+^
SD (ms)	14.35 ± 2.84	14.49 ± 2.88	14.61 ± 2.85	14.35 ± 3.26 *	14.5 ± 3.26	14.54 ± 3.35	15.24 ± 3.22 ^+^
		RMSE	0.56	0.82	0.51	0.54	1.70
CV (%)	1.40 ± 0.30	1.41 ± 0.30	1.43 ± 0.30	1.43 ± 0.33	1.41 ± 0.33	1.42 ± 0.34	1.48 ± 0.33 ^+^
		RMSE	0.06	0.06	0.05	0.05	0.16
DFA (α)	0.76 ± 0.10	0.75 ± 0.10	0.74 ± 0.08	0.76 ± 0.10	0.78 ± 0.10	0.77 ± 0.10	0.72 ± 0.11 ^+^
		RMSE	0.05	0.06	0.04	0.04	0.09
SE (S)	1.58 ± 0.13	1.57 ± 0.15	1.6 ± 0.17	1.56 ± 0.16	1.56 ± 0.13	1.57 ± 0.15	1.65 ± 0.20 ^+^
		RMSE	0.14	0.10	0.08	0.09	0.16

**Table 2 sensors-24-03378-t002:** Level of absolute agreement (two-way agreement intraclass correlation) between the force plates and each sensor. Agreement was classified as poor (<0.5), moderate (0.5–0.75), good (0.75–0.9), or excellent (>0.9). Vicon Blue Trident (BT), Axivity (AX), Xsens Dot (XS), Plantiga (PG), Opal APDM (OP).

	BT	AX	XS	PG	OP
SD	0.98	0.96	0.98	0.98	0.85
CV	0.98	0.94	0.97	0.97	0.88
DFA	0.80	0.82	0.92	0.89	0.67
SE	0.63	0.77	0.80	0.79	0.57

**Table 3 sensors-24-03378-t003:** Level of relative agreement (Pearson’s correlation coefficient) between the force plates and each sensor. Agreement was classified as poor (<0.5), moderate (0.5–0.75), good (0.75–0.9) or excellent (>0.9). Vicon Blue Trident (BT), Axivity (AX), Xsens Dot (XS), Plantiga (PG), Opal APDM (OP).

	BT	AX	XS	PG	OP
SD	0.98	0.97	0.99	0.99	0.88
CV	0.98	0.98	0.97	0.99	0.90
DFA	0.81	0.89	0.91	0.90	0.71
SE	0.65	0.78	0.80	0.80	0.66

## Data Availability

Data is contained within the article.

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
