# Peer review of "The Agreement between Wearable Sensors and Force Plates for the Analysis of Stride Time Variability"

_sensors, 2024, doi:10.3390/s24113378_

Round 1

Reviewer 1 Report

Comments and Suggestions for Authors

The study assesses the agreement between wearable sensors and force plates in measuring stride time variability during military load carriage. It examines linear and non-linear stride variability measures on nineteen Australian Army trainee soldiers. The wearable sensors showed good agreement with force plates, suggesting their viability for remote monitoring in military contexts - an affordable solution for assessing musculoskeletal injury risk. The contributions of the work are significant and fair; however, the manuscript needs a few major corrections before acceptance. 

1. The authors are suggested to expand the validation scope to include additional gait parameters such as stride length and walking speed, which could provide a more comprehensive analysis of the sensors' capabilities.

2. More descriptions and justifications for the specific locations of each sensor placement on the body are needed, as placement can significantly influence data accuracy. It would be helpful if the authors included labels or a legend that details the equipment shown in Figure 1, particularly pointing out the locations of the force plates and any visible sensors.

3. The authors are suggested to compare sensor performance under varying load conditions. This could help understand how changes in load impact the accuracy of wearable sensors. Moreover, a longitudinal analysis is needed to investigate the consistency of wearable sensors over extended periods, which is crucial for military training monitoring.

4. The authors should tell what algorithms are used for data extraction and analysis from the sensors. There should be enough details on the impact of different sampling rates used by the sensors and force plates on the measurements, particularly for non-linear measures.

5. For Figure 3, the authors should discuss the limits of agreement and any outliers visible in the plots, which would improve the reader’s understanding of the data variability and sensor accuracy.

6. At present, results are limited and have minimal explanations. The authors should explore the influence of external factors such as terrain and weather on the sensor data, which are relevant in military settings.

7. Figure 4 is not referred to in the text.  It would be helpful if, in Figure 4, the authors discuss the implications of the broader limits of agreement seen here compared to the linear measures. How are non-linear measures more sensitive to subtle gait changes, as evidenced by the plots?

Reviewer 2 Report

Comments and Suggestions for Authors

The accuracy and validity of inertial measurement units as clinical tools for recording and assessing human locomotion have been intensively investigated recently. They offer substantial advantages compared to the golden standard equipment for this purpose. The study authors attempted to examine the correlation between 5 different wearable sensors and force plates in measuring linear and non-linear stride-time variables during military load carriage to determine their potential utility in assessing musculoskeletal injuries in that field.  

They did so in a clear and well-structured manuscript. The introduction adequately highlights the study's importance and offers background knowledge. It provides the basis for recognizing the research purpose and its related variables.

The methods section is thorough and concisely summarizes the experimental design and procedures. The data analysis and processing techniques are described in a detailed manner, which is crucial for ensuring reproducibility.

However, I would expect a brief explanation of why the authors chose to include these specific 5 types of IMUs. Are they considered the most accurate? Are they the most popular? Etc.

The power analysis, which determined the necessary sample size of 19 subjects to achieve the required statistical power, should be included in the subjects’ part of the methods.

The results section presents a thorough overview of the findings, containing statistical analysis and comparisons between sensors and force plates. It clearly displays both absolute and relative metrics of agreement, with Bland-Altman charts being a useful additional tool to evaluate the level of agreement.

The discussion section is well-structured and effectively analyzes the findings and discusses them within the framework of the current research in the field.

It would be useful to include a paragraph discussing how the lack of arm swing (an essential component of locomotion for human walking) in the current research setup affects the research findings.

Overall, after some minor revisions, I think the manuscript meets the quality requirements and fits the journal's aim and scope.

Round 2

Reviewer 1 Report

Comments and Suggestions for Authors

The authors have addressed all the concerns raised by the reviewer and the manuscript is in better shape. The manuscript could be accepted.